



# The IASI $NH_3$ version 4 product: averaging kernels and improved consistency

Lieven Clarisse[1], Bruno Franco[1], Martin Van Damme[1,2], Tommaso Di Gioacchino[1],
Juliette Hadji-Lazaro[3], Simon Whitburn[1], Lara Noppen[1], Daniel Hurtmans[1], Cathy Clerbaux[3,1], and
Pierre Coheur[1]

[1]Université libre de Bruxelles (ULB), Spectroscopy, Quantum Chemistry and Atmospheric Remote Sensing, Brussels,
Belgium
[2]Royal Belgian Institute for Space Aeronomy, Brussels, Belgium
[3]LATMOS/IPSL, Sorbonne Université, UVSQ, CNRS, Paris, France

**Correspondence:** Lieven Clarisse (Lieven.Clarisse@ulb.be)

**Abstract.** Satellite measurements play an increasingly important role in the study of atmospheric ammonia ($NH_3$). Here, we present version 4 of the Artificial Neural Network for IASI (ANNI) retrieval of $NH_3$. The main change is the introduction of total column averaging kernels (AVKs), which can be used to undo the effect of the vertical profile shape assumption of the retrieval. While the main equations can be matched term for term with analogous ones used in UV/Vis retrievals for other

minor absorbers, we derive the formalism from the ground up, as its applicability to thermal infrared measurements is non-trivial. A large number of other smaller changes were introduced in ANNI v4, most of which improve the consistency of the measurements, across time and across the series of IASI instruments. This includes a more robust way of calculating the hyperspectral range index (HRI), explicitly accounting for long-term changes in $CO_2$ in the HRI calculation and the use of a reprocessed cloud product that was specifically developed for climate applications. The $NH_3$ distributions derived with ANNI

v4 are very similar to the ones derived with v3, although values are about 15–20 % larger due to the improved setup of the HRI. We exclude further large biases of the same nature, by showing the consistency between ANNI v4 derived $NH_3$ columns with columns obtained with an optimal estimation approach. Finally, with v4, we revised the uncertainty budget and now report systematic uncertainty estimates alongside random uncertainties, allowing realistic mean uncertainties to be estimated.

## 1   Introduction

Atmospheric ammonia ($NH_3$) primarily originates from agriculture and related activities. Its presence in the atmosphere leads to a reduction of life quality and to millions of premature deaths via its contribution to particulate matter (Pozzer et al., 2017). As one of the main forms of reactive nitrogen (Galloway et al., 2021), $NH_3$ is also a key element in the global nitrogen cycle with devastating effects on the environment when deposited in excess (Sutton et al., 2014).

Satellite measurements of $NH_3$ abundances have in the past decade contributed to our understanding of its global distri-
bution, spatio-temporal variations, emission sources, concentration trends, transport patterns, chemistry and deposition levels. Currently, the two most widely-used satellite datasets are those derived from observations of the Cross-track Infrared Sounder



(CrIS) (Shephard et al., 2020) and the three Infrared Atmospheric Sounding Interferometers (IASI) (Van Damme et al., 2021). The CrIS product relies on optimal estimation, while the IASI product is based on the conversion of a spectral $NH_3$ index to a total column.

The first version of the IASI-$NH_3$ product (Van Damme et al., 2014) used look-up-tables (LUT) for the conversion. The LUTs were replaced with a more flexible neural network (NN) in Whitburn et al. (2016). Since then, the Artificial Neural Network for IASI (ANNI) retrieval approach underwent a series of incremental improvements that are documented in Van Damme et al. (2017); Franco et al. (2018) and Van Damme et al. (2021). In Franco et al. (2018, 2019, 2020, 2022) and Rosanka et al. (2021) the ANNI retrieval framework was expanded for the retrieval of other minor trace gases from IASI observations. A
similar retrieval approach was also recently adopted for isoprene retrievals from CrIS (Wells et al., 2022).

    Returning to ANNI-$NH_3$, in Van Damme et al. (2017), a *reanalysis product* was introduced. This product differs in the origin of the input parameters that are used for the NN. Whereas the baseline product (also called near-real time (NRT) product) uses operational IASI Level 2 (L2) information on the pressure, humidity and temperature profiles and a climatology characterizing the boundary layer height, the reanalysed product uses ERA5 model output for these parameters (Hersbach et al., 2020),
interpolated in time and space to match the observations. The resulting product is temporally more consistent as it removes the effect of the several changes that occurred in the L2 products throughout the years. Both $NH_3$ products include several empirical corrections to counter small differences observed between the three IASI instruments and small biases that occurred as a result of sporadic changes to the IASI instrument or in the L0 to L1c processing of the spectra.

    The IASI $NH_3$ product is widely used by the scientific community. However, the absence of averaging kernels (AVKs) has
in the past hampered model comparison and assimilation. This paper presents version 4 of the ANNI retrieval framework, with the most important change being the introduction of AVKs. After a brief recapitulation of the retrieval algorithm in Sect. 2, the AVK framework is presented in Sect. 3. This includes its theoretical basis, practicalities related to how the AVKs are calculated within ANNI and a discussion on how they can be used in measurement-model comparison and assimilation. Other changes that were introduced in ANNI v4 are detailed in Sect. 4 (for those related to temporal consistency) and Sect. 5 (for all other
changes). In Sect. 6, an evaluation of the $NH_3$ product is presented, comparing the v3 with the v4 product and the neural network output with retrievals based on optimal estimation. In the final part of this paper, we present the revised uncertainty budget of the ANNI retrieval.

## 2   ANNI retrieval overview

Here, we give a brief overview of the ANNI algorithm, and refer to the previously cited papers on the $NH_3$ product for a
detailed description and the rationale behind the different retrieval choices. Table 1 summarizes the most important quantities and symbols used in this paper.

    The retrieval consists of two independent computational steps. The first one characterizes the $NH_3$ signal strength in a spectrum $\boldsymbol{L}$, via the so-called hyperspectral range index (HRI), which relies on a mean spectrum $\bar{\boldsymbol{L}}$ and associated covariance





matrix $\mathbf{S}$ constructed from a set of spectra with no observable $NH_3$ spectral signatures. It is defined as

$$\text{HRI} = N\boldsymbol{K}^{\text{T}}\mathbf{S}^{-1}(\boldsymbol{L} - \bar{\boldsymbol{L}}), \tag{1}$$

with $N$ a normalization constant and $\boldsymbol{K}$ an $NH_3$ Jacobian. By construction, the HRI has a mean of zero on the spectra from which $\bar{\boldsymbol{L}}$ and $\mathbf{S}$ are constructed. The normalization factor guarantees that the HRI has a standard deviation of one on spectra containing only background levels of $NH_3$.

The second part of the algorithm relies on a neural network to link the measured HRIs to estimates $\hat{X}^a$ of the true $NH_3$ total columns $X$, via a scaling factor $\text{SF}^a$:

$$\hat{X}^a = \frac{\text{HRI}}{\text{SF}^a} + B, \tag{2}$$

with $B$ an $NH_3$ background column. In the past, the background column was always assumed to be zero for $NH_3$. In what follows, we assume an arbitrary background column $B$, so that the recipe can be applied to the other tracers retrieved from IASI with ANNI (e.g. $CH_3OH$ or PAN). Retrieved quantities will be indicated with a hat, as in $\hat{X}$. Superscripts refer to the assumed or modelled vertical profile shape. In the ANNI retrieval framework, the scaling factor $\text{SF}^a$ is the quantity that is calculated with a NN, for each individual observation, based on the state of the atmosphere (temperature and water vapour profile, surface pressure), the surface temperature and emissivity, the zenith angle, the HRI and an assumed vertical profile shape. For $NH_3$, the volume mixing ratio (VMR) vertical profile is modelled as a Gaussian

$$\text{VMR}(z) = \text{VMR}(z_0)e^{-\frac{(z-z_0)^2}{2\sigma^2}}, \tag{3}$$

with $z$ the altitude about ground level, $z_0$ the peak altitude and $\sigma$ the width of the profile. Over land, the peak altitude is set at the surface, with a width $\sigma$ equal to the boundary layer height. Over ocean, the peak altitude is set to 1.4 km with a $\sigma$ of 0.9 km. In general $\hat{X}^a \neq X$ because of instrumental noise, errors in the assumed vertical profile, imperfect knowledge of one of the other input parameters and errors in the spectroscopic parameters or forward model. The NN is trained from a large set of forward modelled spectra. Appropriate pre- and post-retrieval flags accompany the retrieval. The pre-filter removes respectively measurements with erroneous L1 or excess cloud coverage. The post-filter flags retrievals with limited or no sensitivity to the measured quantity

$$\frac{1}{|\text{SF}|} > 1.5 \cdot 10^{16} \text{molec.cm}^{-2} \tag{4}$$

and retrievals whose HRIs are either too noisy or for which the assumed vertical profile is incompatible with the measured HRI

$$|\text{HRI}| > 1.5 \quad \text{and} \quad \hat{X}^a < 0. \tag{5}$$

Via propagation of uncertainty, a total retrieval uncertainty is calculated for each individual measurement alongside the retrieved column.



## 3 Averaging kernels

The general AVK framework that we introduce below, bears a lot of similarity to the total column AVK formalism (Eskes and Boersma, 2003) developed for the DOAS retrieval approach of weakly absorbing species (see also Palmer et al. (2001); Rodgers and Connor (2003); Boersma et al. (2004, 2016); Cooper et al. (2020)). In the DOAS retrieval approach, the total column $X$ is retrieved as

$$\hat{X}^a = \frac{\mathrm{SCD}}{\mathrm{AMF}^a}, \tag{6}$$

with SCD, the slant column density and AMF the air mass factor which accounts for the atmospheric conditions and assumed vertical profile. Eq. (6) has the same functional form as the main formula (Eq. (2)) of the ANNI retrieval formalism, with the SCD corresponding to the HRI and the AMF to the SF provided by a NN.

One key element on which the total column AVK formalism of Eskes and Boersma (2003) relies, is linearity and additivity of the spectrum with respect to changes in the trace gas amount. Linearity is a consequence of the curve of growth of spectral lines for low optical depths (Thorne et al., 1999) and in the DOAS approach also implies that the SCD is proportional to the trace gas abundance. Additivity represents the fact that the effect of different atmospheric layers can be summed up independently from each other. Given the definition of the HRI and the effects of thermal emission of the atmosphere, it is not obvious that these hold in the infrared spectral domain, and for this reason, we derive both properties below.

### 3.1 On the linearity and additivity of the HRI

Let $L_\nu^B$ be the radiance at the sensor for a scene with climatological background levels $B$ of the target trace gas and corresponding HRI of zero. Dividing the atmosphere in $n$ appropriately spaced layers $z$, we denote by $B_z$ the corresponding partial columns ($B = \sum_z B_z$). We can then calculate $L_\nu^B$ from the sequence of equations (Rodgers, 2000; Petty, 2006)

$$L_{1\nu}^B = t_{1\nu}(L_{0\nu} - B_{1\nu}) + B_{1\nu} \tag{7}$$

$$L_{2\nu}^B = t_{2\nu}(L_{1\nu} - B_{2\nu}) + B_{2\nu} \tag{8}$$

$$\vdots$$

$$L_\nu^B = t_{n\nu}(L_{(n-1)\nu} - B_{n\nu}) + B_{n\nu}. \tag{9}$$

Forward substitution of the above relations yields $L_\nu^B$ as a function of the surface term $L_{0\nu}$. Here $t_{z\nu}$ are the layer transmittances that account for the absorption of all atmospheric species. The $z$ dependence of this parameter is related to vertical variations in the atmospheric constituents and the pressure and temperature dependence of the line intensities. $B_{z\nu} = B_\nu(T_z)$ corresponds to the Planck's blackbody function for an averaged layer temperature $T_z$.

For this scene, we now introduce an additional trace amount $X - B$ and write the corresponding observed radiance as $L_\nu$. In each layer, the transmittance will decrease by a factor $t_{z\nu}^{X_z - B_z} = e^{-\tau_{z\nu}} \approx 1 - \tau_{z\nu}$, with $\tau_{z\nu}$ the small optical depth caused by the excess $X_z - B_z$. With this, the sequence of equations becomes

$$L_{1\nu} = t_{1\nu}(1 - \tau_{1\nu})(L_{0\nu} - B_{1\nu}) + B_{1\nu} \tag{10}$$





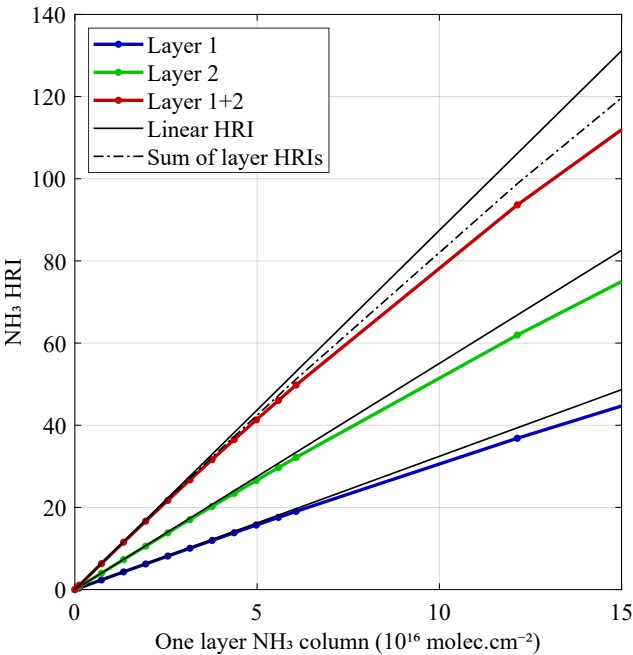

**Figure 1.** Numerical demonstration of the linearity and additivity of the HRI as a function of a change in partial column. In the blue and green scenario, $NH_3$ was varied at one fixed altitude. In the red scenario, partial columns in both layers were varied simultaneously. The black solid lines represent linearity, whereas the dash dotted line, being the sum of the green and blue curve, represents additivity.

$$L_{2\nu} = t_{2\nu}(1 - \tau_{2\nu})(L_{1\nu} - B_{2\nu}) + B_{2\nu} \tag{11}$$

$$\vdots$$

$$L_\nu = t_{n\nu}(1 - \tau_{n\nu})(L_{(n-1)\nu} - B_{n\nu}) + B_{n\nu}. \tag{12}$$

For optical thicknesses of the target trace gas $\tau_{z\nu}$ well below one, the second ($\tau_{i\nu}\tau_{j\nu}$) and higher order terms in optical depth can be neglected. Combining both set of equations, one can verify that

$$L_{1\nu} = L_{1\nu}^B - \tau_{1\nu}(L_{0\nu} - B_{1\nu})t_{1\nu} \tag{13}$$

$$L_{2\nu} = L_{2\nu}^B - \tau_{1\nu}(L_{0\nu} - B_{1\nu})t_{1\nu}t_{2\nu} - \tau_{2\nu}(L_{1\nu} - B_{2\nu})t_{2\nu} \tag{14}$$

$$\vdots$$

$$L_\nu = L_\nu^B - \sum_{z=1}^n \tau_{z\nu}(L_{(z-1)\nu}^B - B_{z\nu})\prod_{j=z}^n t_{j\nu}. \tag{15}$$

Each of the terms in the sum express the effect of the absorption in one layer due to the excess $X_z - B_z$ and attenuated by the layers above. Note that the absorptions are proportional to both the optical depth and the local thermal contrast ($L_{(z-1)\nu}^B - B_{z\nu}$), which are two parameters that drive the measurement sensitivity in the infrared (Bauduin et al., 2017). The optical thickness in



turn is proportional to the partial column of the target species $\tau_{z\nu} \propto (X_z - B_z)$ and thus

$$L_\nu = L_\nu^B - \sum_z c_{z\nu}(X_z - B_z). \tag{16}$$

The constants $c_{z\nu}$ depend on the state of the atmosphere at level $z$ and above, but are independent of the excess trace gas amount $X - B$.

The HRI is by definition a linear combination of spectral channels $\nu$, from Eq. (1)

$$\mathrm{HRI} = \sum_\nu w_\nu L_\nu + C, \tag{17}$$

with $C$ and $w_\nu$ numerical constants. Using Eq. (16) and the fact that the HRI is zero on the background ($\mathrm{HRI} = \sum_\nu w_\nu L_\nu^B + C = 0$), we find

$$\mathrm{HRI} = \sum_\nu w_\nu L_\nu^B - \sum_{\nu,z} w_\nu c_{z\nu}(X_z - B_z) + C \tag{18}$$

$$= -\sum_{\nu,z} w_\nu c_{z\nu}(X_z - B_z). \tag{19}$$

This equation implies that for optically thin absorbers, the HRI can be written as a weighted sum of the partial column enhancements. This linearity and additivity of the $NH_3$ HRI as a function of partial columns is illustrated in Fig. 1 by means of simulations with a radiative transfer code. The line in blue illustrates the nearly linear increase in HRI for increasing $NH_3$ columns in the first atmospheric layer of the model. Starting from about $7 \cdot 10^{16}$ molec.cm$^{-1}$, slow departure from linearity is observed. The green line shows the same for the second atmospheric layer. The overall larger HRI in this second layer results

from higher thermal contrast (TC) higher up in the atmosphere. Finally, the red line represents the HRI when $NH_3$ is increased simultaneously in both layers. The dash-dotted line is the sum of the blue and green curve and for low columns is almost indiscernible from the simulated HRI, illustrating additivity in the optically thin limit.

To make the link with Eq. (2), Eq. (19) can be rewritten as

$$\mathrm{HRI} = \sum_z \mathrm{SF}_z(X_z - B_z) \tag{20}$$

$$= \sum_z \mathrm{SF}_z \frac{X_z - B_z}{X - B}(X - B) \tag{21}$$

$$= \mathrm{SF}(X - B), \tag{22}$$

with $\mathrm{SF}_z$ local scaling factors and

$$\mathrm{SF} = \sum_z \mathrm{SF}_z \frac{X_z - B_z}{X - B}, \tag{23}$$

total scaling factors that depend on the scene conditions (e.g. surface temperature, atmospheric temperature and pressure

profiles, vertical profile of the target species). We note that whereas SF depends on the normalized vertical profile shape $\frac{X_z - B_z}{X - B}$, both $\mathrm{SF}_z$ and SF are independent of the total column for a fixed profile shape.



Finally, introducing

$$\text{HRI}_z = \text{SF}_z(X_z - B_z), \tag{24}$$

we see from Eq. (20) that the HRI can be decomposed in different partial $\text{HRI}_z$

$$\text{HRI} = \sum_z \text{SF}_z(X_z - B_z) = \sum_z \text{HRI}_z. \tag{25}$$

These $\text{HRI}_z$ quantify how much each layer contributes to the total HRI and they are, again in the optically thin limit, independent from each other.

### 3.2 Total column averaging kernels

Equation (22) motivates the NN retrieval formula

$$\hat{X}^a = \frac{\text{HRI}}{\text{SF}^a} + B, \tag{26}$$

with $\text{SF}^a$ an estimated scaling factor which depends on the best estimates of all the dependencies of SF, including the vertical profile of the target species. As we have just shown, in the optically thin limit $\text{SF}^a$ is independent of the total column amount. This is no longer the case for strong absorptions when non-linear effects become increasingly important. In the ANNI retrieval framework this is taken care off by including the HRI as an input parameter in the calculation of the SF. In what follows, we will first derive the AVK formalism in the optically thin limit, assuming both linearity and additivity (and therefore SF that are independent of the (partial) column). In Sect. 3.5 we will then show how to correct for small errors that arise when these conditions are not met.

The quantities $\hat{X}_z - B_z$ express the local enhancement of the trace gas column at an altitude $z$, and it is their relative proportions that are constrained in the retrieval by the use of an a priori profile shape. We write

$$a_z = \frac{\hat{X}_z^a - B_z}{\hat{X}^a - B}, \tag{27}$$

for the normalized a-priori vertical partial column profile, or partial column profile shape, with $\hat{X}_z^a$ the partial columns corresponding to the retrieved $\hat{X}^a$ and $\sum_z a_z = 1$. In the previous section we demonstrated that in the optically thin limit, the scaling factor $\text{SF}^a$ equals the weighted sum of the different local scaling factors defined by the profile shape:

$$\text{SF}^a = \sum_z \text{SF}_z \frac{X_z^a - B_z}{X^a - B} = \sum_z \text{SF}_z a_z. \tag{28}$$

Defining the averaging kernel as

$$A_z^a = \frac{\text{SF}_z}{\text{SF}^a} = \frac{\text{SF}_z}{\sum_z \text{SF}_z a_z}, \tag{29}$$

we can express $\hat{X}^a$ as a function of $X_z$ by eliminating the HRI from Eq. (20) and Eq. (26):

$$\hat{X}^a = \frac{\text{HRI}}{\text{SF}^a} + B \tag{30}$$





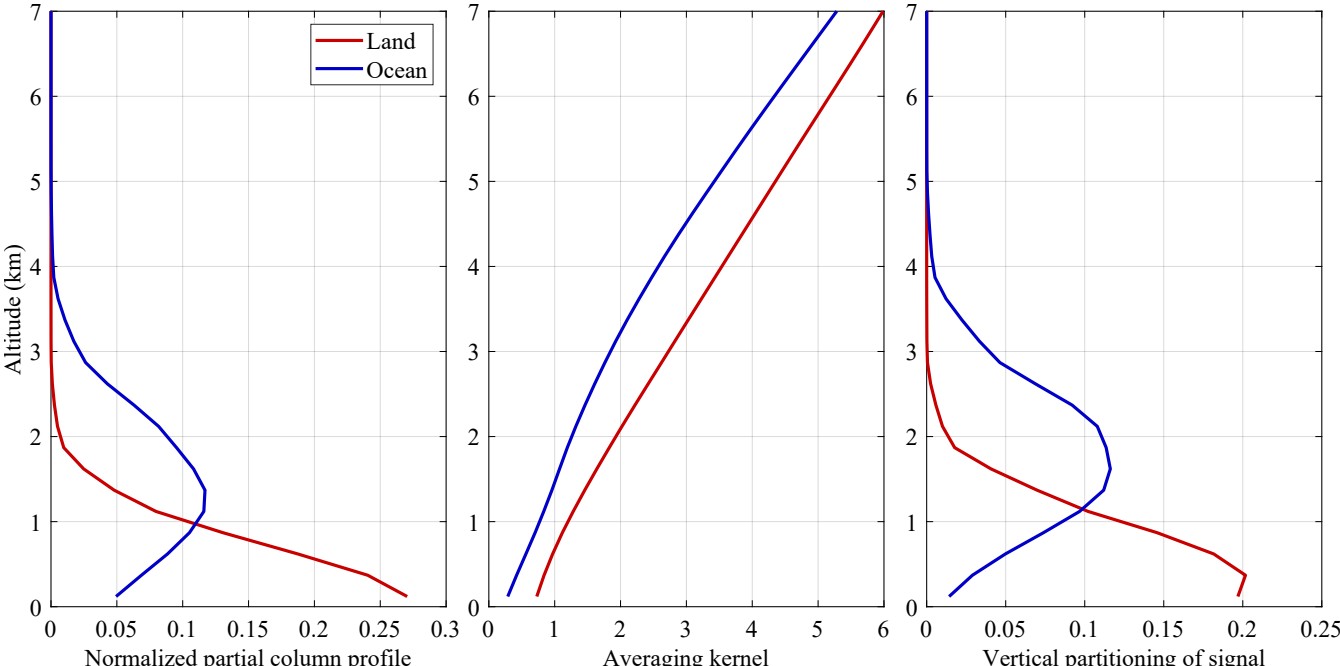

**Figure 2.** A priori vertical profile $a_z$, averaging kernel $A_z^a$ and vertical partitioning of signal $V_z^a$.

$$= \frac{\sum_z \mathrm{SF}_z (X_z - B_z)}{\mathrm{SF}^a} + B \tag{31}$$

$$= \sum_z A_z^a (X_z - B_z) + B. \tag{32}$$

As can be seen from this equation, the averaging kernel $A_z^a$ fully characterizes the measurement, and can be used to mathematically map the true profile $X_z$ to the measured total column $\hat{X}^a$.

Two example AVKs for an $NH_3$ retrieval are shown in the middle panel of Fig. 2, for a retrieval over land and over ocean, with a priori profiles shown in the left panel. Both AVKs logically increase with altitude, as the temperature difference between the surface and a given atmospheric layer, and therefore the scaling factor, increases with altitude. Apart from a multiplicative constant $(1/\mathrm{SF}^a)$, the AVKs are independent from the a priori profiles, which explains why both land and ocean AVKs increase similarly with altitude. The multiplicative constant determines the altitude for which the AVK is one. This altitude can be interpreted as an equivalent effective $NH_3$ altitude, where all $NH_3$ can be thought to be located at. For the ocean and land retrieval, this altitude is located respectively around 1.6 and 0.7 km, consistent with the a priori profiles shown in the left panel.





**Table 1.** Summary of the main quantities and associated symbols that are used in the AVK formalism.

| Name | Symbol | Notes/variations |
|---|---|---|
| Total column | $X$ | $\hat{X}^a, \hat{X}^m$ (retrieved with $a$ or $m$ profile); $M^m, \hat{M}^a$ (modelled or retrieved-modelled profile) |
| Partial column | $X_z$ | $X = \sum X_z$ |
| Confined column | $\hat{X}^{\|z}$ | retrieval assuming the entire total column is localized at altitude $z$ |
| Background column | $B$ | $B = \sum B_z$ |
| Normalized a priori profile | $a_z$ | $a_z = (\hat{X}_z^a - B_z)/(\hat{X}^a - B)$ |
| Normalized modelled profile | $m_z$ | $m_z = (M_z^m - B_z)/(M^m - B)$ |
| Total scaling factor | SF | $\text{SF}^a, \text{SF}^{\|z}$ (a priori or confined profile) |
| Local scaling factor | $\text{SF}_z$ | $\text{SF} = \sum_z \text{SF}_z(X_z - B_z)/(X - B)$, $\text{SF}^a = \sum_z \text{SF}_z a_z$ |
| Averaging kernel | $A_z^a$ | $A_z^a = \text{SF}_z/\text{SF}^a$ |
| Vertical partitioning of signal | $V_z^a$ | $V_z^a = A_z^a a_z$ with $\sum V_z^a = 1$ |

## 3.3 Interpretation

It is instructive to compare the total column averaging kernel, as defined above, with the one arising in optimal estimation retrievals (Rodgers, 2000):

$$\hat{X}^a = (\mathbf{I} - \mathbf{A})X^a + \mathbf{A}X. \tag{33}$$

Here, we use matrix and vector notation, with $X$ and $\hat{X}$ respectively the true and retrieved partial columns and $\mathbf{A}$ the AVK matrix. $\mathbf{I}$ is the identity matrix. Both Eq. (32) and Eq. (33) allow simulating the retrieval process for any hypothetical $X$ (e.g. from a model, or an independent measurement). However, this is largely where their similarity ends, as there are important differences when it comes to interpreting these two types of AVK.

The first is the role of the a priori. For the total column retrieval, the a priori fixes only the vertical profile shape, while for the optimal estimation retrieval, the a priori affects both the vertical profile shape, and the retrieved value at each altitude separately. Eq. (33) expresses that the retrieved profile is a weighted sum of the a priori and the true profile, with the weights provided by the AVK. When the information content of the measurement is low or the retrieval is too heavily constrained, the AVK will tend toward zero and the solution will remain close to the a priori. Conversely, when the information content is high or the retrieval loosely constrained, the AVK will approach the unit matrix. The optimal estimation AVK is therefore a measure of how much information is extracted from the measurement, with its trace commonly denoted 'degrees of freedom for signal'. By contrast, the total column averaging kernels introduced above, are not a measure of how much information is extracted from the measurement, as they accompany an unconstrained retrieval. A perfect measurement would correspond to an all-ones vector $A_z = 1$ for all $z$. However, inherent to (infrared) sounding, sensitivity varies as a function of thermal contrast, and thus altitude, so that this ideal can never be met.



The second important difference relates to the fact that a vertical profile is retrieved in the optimal estimation type retrievals. The rows of $\mathbf{A}$ express the vertical resolution of the retrieval, where the ideal is a narrow function that peaks at its corresponding altitude. As no vertical information is extracted in the total column retrievals, this interpretation also does not apply. $A_z$ is a (normalized) vector that shows which layers of the atmosphere offer *in principle* the greatest sensitivity. It naturally peaks high up in the atmosphere, irrespective of the location of the trace gas. However, combining the total column AVK with the a priori profile, does allow extracting a vector which characterizes the vertical sensitivity, as we now show. Starting from Eq. (26) and Eq. (28) we have

$$\text{HRI} = \text{SF}^a(\hat{X}^a - B) \tag{34}$$

$$= \sum_z \text{SF}_z(\hat{X}^a - B)a_z \tag{35}$$

$$= \sum_z \text{SF}_z(\hat{X}_z^a - B_z) \tag{36}$$

$$= \sum_z \widehat{\text{HRI}}_z^a, \tag{37}$$

where we defined $\widehat{\text{HRI}}_z^a = \text{SF}_z(\hat{X}_z^a - B_z)$, similarly to Eq. (24) where we defined $\text{HRI}_z = \text{SF}_z(X_z - B_z)$. A retrieval that assumes a given a priori vertical profile $a_z$, therefore implicitly assumes that the HRI (the trace gas signal) can be decomposed into partial $\widehat{\text{HRI}}_z^a$ corresponding to spectral change at each altitude. Note that while $\text{HRI} = \sum_z \widehat{\text{HRI}}_z^a = \sum_z \text{HRI}_z$, the same does not necessarily hold for each individual level $\widehat{\text{HRI}}_z^a \neq \text{HRI}_z$, as the assumed profile can differ from the actual profile. With this we can define the normalized *assumed* HRI profile or equivalently, the *probable* vertical partitioning $V_z^a$ of the signal

$$V_z^a = \frac{\widehat{\text{HRI}}_z^a}{\text{HRI}}, = \frac{\text{SF}_z(\hat{X}_z^a - B_z)}{\text{SF}^a(\hat{X}^a - B)} = A_z^a a_z. \tag{38}$$

From the first equality follows that $\sum_z V_z^a = 1$ (also follows the last term and Eq. (29)). The three profiles $a_z$, $A_z^a$ and $V_z^a$ are illustrated in Fig. 2 for a typical $NH_3$ retrieval. As can be seen, the maximum of $V_z^a$ is shifted upwards compared to $a_z$, due to the more favourable thermal contrast higher up in the atmosphere.

## 3.4 AVK application

There are two alternative ways in which averaging kernels can be exploited to remove the impact of the vertical profile assumption of the retrieval (Palmer et al., 2001; Eskes and Boersma, 2003; Cooper et al., 2020).

### 3.4.1 Method 1: Simulating measurements of the modelled columns

Let $M_z^m$ be a modelled profile with corresponding total column $M^m$ and normalized profile (enhancement)

$$m_z = \frac{M_z^m - B_z}{M^m - B}. \tag{39}$$

We can simulate what would have been retrieved if the modelled profiles were observed using Eq. (32):

$$\hat{M}^a = \sum_z A_z^a(M_z^m - B_z) + B. \tag{40}$$



This $\hat{M}^a$ is directly comparable with $\hat{X}^a$ as the same a priori profile shape $a_z$ is used for both retrievals. However, in case the a priori profile significantly differs from the truth, both $\hat{X}^a$ and $\hat{M}^a$ can deviate far from the truth.

### 3.4.2 Method 2: Using modelled vertical profiles as a priori

240 Rather than altering the modelled column, an attractive alternative is to alter the retrieved column to use instead of the a priori vertical profile, the modelled profile (see Boersma et al. (2016), Appendix D)

$$\hat{X}^m = \frac{\text{HRI}}{\text{SF}^m} + B \tag{41}$$

$$= \frac{\text{SF}^a(\hat{X}^a - B)}{\sum_z \text{SF}_z m_z} + B \tag{42}$$

$$= \frac{\hat{X}^a - B}{\sum_z A_z^a m_z} + B. \tag{43}$$

245 The averaging kernel associated with $\hat{X}^m$ is

$$A_z^m = \frac{\text{SF}_z}{\text{SF}^m} = A_z^a \frac{\text{SF}^a}{\text{SF}^m}. \tag{44}$$

This $\hat{X}^m$ can be directly compared with $M^m$, as both employ the same profile $m_z$. Note that $\hat{X}^m$ depends only on the shape of the modelled profile, not the total column. This method can be used to obtain an improved retrieval by using a modelled profile that approaches the reality better than a static a priori profile. Since it was first applied to $NH_3$ (Whitburn et al., 2016), 250 the ANNI retrieval has been capable of using modelled profiles, by adapting the input parameters to the network. However, when the modelled profiles were changed, the entire retrieval process had to be redone. Using Eq. (43) and the provided AVKs, changing the a priori profile can be done a posteriori by the data users. An important pracial note is that the post-filter of the retrieval (see Sect. 2) includes a threshold on the scaling factor, and that this filter should be reevaluated for $\hat{X}^m$ using $\text{SF}^m$.

Both methods can be summarized as

255 $$M^m \xrightarrow{\text{method 1}} \hat{M}^a \quad \text{to be compared with} \quad \hat{X}^a \tag{45}$$

$$\hat{X}^a \xrightarrow{\text{method 2}} \hat{X}^m \quad \text{to be compared with} \quad M^m. \tag{46}$$

Eq. (39), (40) and (43) can be combined as

$$\hat{X}^m = \frac{(\hat{X}^a - B)(M^m - B)}{\sum_z A_z^a(M_z^m - B_z)} + B \tag{47}$$

$$= \frac{(\hat{X}^a - B)(M^m - B)}{\hat{M}^a - B} + B, \tag{48}$$

260 or

$$\frac{\hat{X}^m - B}{M^m - B} = \frac{\hat{X}^a - B}{\hat{M}^a - B}, \tag{49}$$

which shows that, when the goal is to compare the ratio between model and retrieved columns, both methods are equivalent (Cooper et al., 2020).





### 3.5 Practical considerations

In the ANNI retrieval formalism, the total scaling factor $SF^a$ is calculated directly by the NN, and not calculated via intermediate $SF_z$ and application of Eq. (28). These $SF_z$, which are required to calculate the AVKs, can however be estimated by exploiting the flexibility of the NN. For $NH_3$, the NN is trained for a wide variety of Gaussian profiles, with peak altitudes ranging from 0 to 20 km and $\sigma$ from 100 m to 3 km. The $SF_z$ can be estimated from the network using the input parameters $z_{peak} = z$ and $\sigma = 100$ m for the Gaussian profile. For this calculation, an HRI input parameter is also required and the choice was made to use the observed HRI. The corresponding column that can be calculated from this satisfies

$$\hat{X}^{|z} = \frac{HRI}{SF^{|z}} + B, \tag{50}$$

where $\hat{X}^{|z}$ is the retrieved total column assuming all the trace gas enhancement is situated in the narrow Gaussian band around altitude $z$. $SF^{|z}$ is the corresponding total scaling factor, which is used to approximate the local scaling factor $SF_z$. With this the AVK can be constructed as

$$A_z^a = \frac{SF_z}{SF^a} \approx \frac{SF^{|z}}{SF^a} = \frac{(\hat{X}^a - B)HRI}{(\hat{X}^{|z} - B)HRI} = \frac{\hat{X}^a - B}{\hat{X}^{|z} - B}. \tag{51}$$

The formulas provided above are exact in the linear limit, but for large columns, $SF^a$ and $SF^{|z}$ have an increasingly high dependence on the value of the HRI. The NN takes into account this dependence so that Eq. (2) and Eq. (50) are always good approximations of the true $X$, provided that either the assumed a priori profile is correct or that the tracer is confined to a narrow layer. However, there is no guarantee that

$$SF^a \stackrel{?}{=} \sum_z SF_z a_z, \tag{52}$$

or thus that $\sum_z A_z^a a_z = N$ equals one. A consistent AVK can however be obtained as

$$A_z^a = \frac{1}{N} \frac{\hat{X}^a - B}{\hat{X}^{|z} - B}. \tag{53}$$

This normalization also guarantees that applying the averaging kernels on the a priori vertical profile returns back the retrieved column:

$$\sum_z A_z^a(\hat{X}_z^a - B_z) + B = \sum_z A_z^a a_z(\hat{X}^a - B) + B = \hat{X}^a. \tag{54}$$

The normalization factors are shown in Fig. 3 for one day of IASI $NH_3$ measurements (morning overpass). Over ocean they average $0.99 \pm 0.05$ and over land $0.98 \pm 0.03$, illustrating the consistency of the approach and the fact that non-linear effects are modest. The areas where the normalization factors are furthest from one seem to be affected by clouds or aerosol (e.g. off the East coast of California and the East coast of North Africa).

The necessity of using normalization factors follows from the fact that in the non-linear regime, the $SF_z$ are not uniquely defined, depending on the concentration in the layer $z$ (non-linearity) and the other layers (non-additivity). However, as we





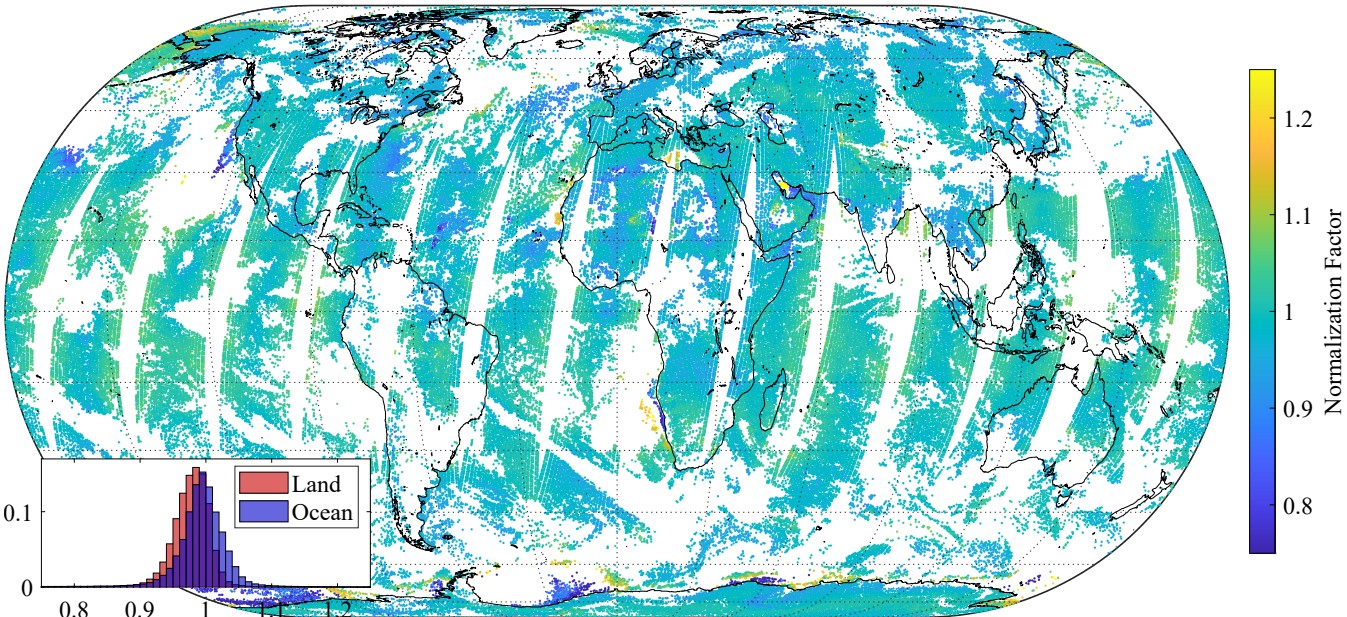

**Figure 3.** AVK normalization factors for one day of IASI observations (17 June 2015, morning overpass of IASI-A for observations with a cloud cover less than 25 %).

have shown, in the neighbourhood of the solution, a fully consistent AVK can be obtained after renormalization. It is this AVK that is recommended when applying method 1 in model comparisons. However, in case method 2 is employed and when the modelled profile concerns a narrow layer at high altitude (e.g. pyro-convective fire plume) or significantly deviates from the a

priori, it can be better not to renormalize. In particular, when we have a narrow modelled profile layer at an altitude $z'$, with $m_z = \delta_{zz'}$, the second method, without renormalization, yields the expected

$$\hat{X}^m = \frac{\hat{X}^a - B}{\sum_z A_z^a \delta_{zz'}} + B \tag{55}$$

$$= \frac{\hat{X}^a - B}{A_{z'}^a} + B \tag{56}$$

$$= (\hat{X}^{|z'} - B) + B \tag{57}$$

$$= \hat{X}^{|z'}. \tag{58}$$

The output files of the ANNI retrieval contain $\hat{X}^a$, $\hat{X}^{|z}$, $B_z$ and $N$. With this, $A_z^a$ can be calculated if required from Eq. (51) or renormalized via Eq. (53) .



## 4 Temporal consistency

### 4.1 Generalized inverse

The generalized error covariance matrix $\mathbf{S}$ plays a key role in the calculation of the HRI. As a symmetric matrix, $\mathbf{S}$ has real eigenvalues $\lambda_i$ and can be decomposed as

$$\mathbf{S} = \sum_{i=1}^{n} \lambda_i \boldsymbol{s_i} \boldsymbol{s_i^T}, \tag{59}$$

with all $\boldsymbol{s_i}$ orthogonal to each other. It follows that its inverse can be written as

$$\mathbf{S}^{-1} = \sum_{i=1}^{n} \frac{1}{\lambda_i} \boldsymbol{s_i} \boldsymbol{s_i^T}. \tag{60}$$

This expression leads to an intuitive interpretation of the HRI (Clarisse et al., 2013): it can be seen as a weighted projection of the spectrum onto the Jacobian, with the directions that usually exhibit the most variability carrying the lowest weight.

The distribution of the eigenvalues of the covariance matrix used for the $NH_3$ HRI (with a spectral range covering 812 to 1126 $cm^{-1}$) is shown in Fig. 4. Three domains can be distinguished: (i) the 30 highest values, corresponding to the principal components, (ii) around 1200 values corresponding mainly to instrumental noise and (iii) around 20 very small eigenvalues.

These smallest eigenvalues are of the order of the numerical precision at which the covariance matrix is calculated, and in essence correspond to directions not occurring in IASI spectra. While random instrumental noise would be expected to occur in all directions, apodization and L1 post-processing remove some. Such directions carry the most weight in $\mathbf{S}^{-1}$, but as they are not found in real spectra they do not contribute much to the total HRI (as can easily be verified numerically).

However, small changes to the instrument calibration or post-processing can alter the contribution of these directions in

the IASI spectra, and because they carry such a large weight in the HRI, they can affect its value considerably. This explains why the HRI in the past has been found to be very sensitive to such changes (Van Damme et al., 2021). It also explains the occurrence of (small) biases between the different instruments. The solution is fortunately simple (Rodgers and Connor, 2003; Eaton, 2007), and is obtained by disregarding the terms corresponding to the very small eigenvalues in Eq. (60). As we will show later, using such a generalized inverse does not eliminate the effects of L1C changes completely, but reduces their

magnitude considerably.

After the generalized inverse was implemented, an unexpected change was observed in the value of the HRIs on spectra from the period on which the covariance matrix was built. It turns out that while the scalar product of observed IASI spectra with the eigenvectors corresponding to the lowest eigenvalues $\boldsymbol{L^T} \boldsymbol{s_i}/\lambda_i$ is near zero, this is not the case for spectra generated with the forward model. This is due to small discrepancies between spectra generated by the forward and actual spectra, that

are magnified by the $1/\lambda_i$ factors. Hence, synthetic HRIs calculated on the training set of the neural network, have in the past been overestimated, resulting in low biases in the retrieved columns. As we will show in Sect. 6, the magnitude of this bias was in ANNI-v3 around $18\%$. For the retrieval of other trace gases presented in Franco et al. (2018), especially those operating on a smaller spectral range, the bias has been evaluated to be much smaller.





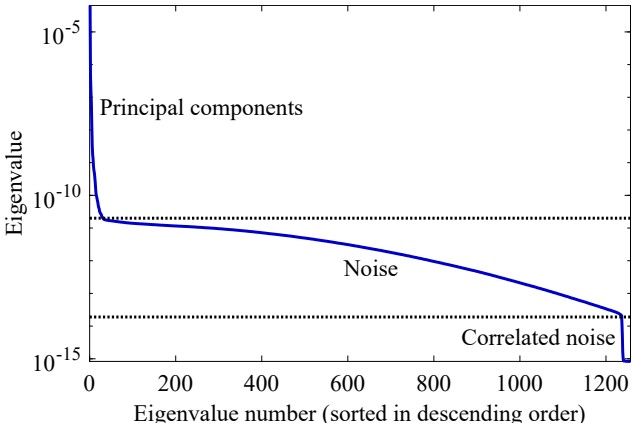

**Figure 4.** Eigenvalue spectrum of the covariance matrix **S** used for the calculation of the HRI of $NH_3$. The eigenvalues are ordered from largest to smallest.

## 4.2 Carbon dioxide

As the mean spectrum and covariance matrix that are used for the HRI are calculated from spectra measured within one reference year (2013), long-term changes in atmospheric composition that affect the spectral region of interest, can have unwanted effects on the HRI. This was first noted in Van Damme et al. (2021), where a spurious trend was seen in the HRI $NH_3$ data over remote regions. It was attributed to the increase in global carbon dioxide ($CO_2$) concentrations, because of the presence of a weak $CO_2$ absorption band in the 920–990 $cm^{-1}$ spectral region (Whitburn et al., 2021) where $NH_3$ has its strongest absorption. A linear correction on the HRI of the order of 0.03 per year was introduced to compensate for this effect. However, because of seasonal variations, and possible temperature dependence of the interference, an HRI which is less sensitive to $CO_2$ changes is preferable. One option is to build the covariance matrix from spectra spanning the entire period of IASI measurements.

An alternative approach is to account directly for the effects of $CO_2$ in the calculation of the HRI. The HRI formula is related to generalized least squares estimation and can be expanded to include multiple variables that are simultaneously estimated (Walker et al., 2011; Theys et al., 2022). In our case, the Jacobian vector becomes a two-column matrix, one column corresponding to $NH_3$ and the other to $CO_2$. The HRI formula Eq. (1) remains formally identical (with only the first component of the two-element HRI vector of interest). The effect of this change on the long-term trend of the HRI is detailed in Sect. 4.4.

## 4.3 Cloud clearing

The ERA5 model output replaces satisfactorily the IASI L2 for all input parameters, except for the surface temperature and cloud cover. These are spatially and temporally too variable for model output to be representative for an IASI footprint at a given time. All previous reanalysed ANNI-$NH_3$ products still relied on different versions of the IASI L2 cloud product. Recently, Whitburn et al. (2022) developed a NN-based cloud flag. Trained with data from the latest version (v6.5) of the official L2





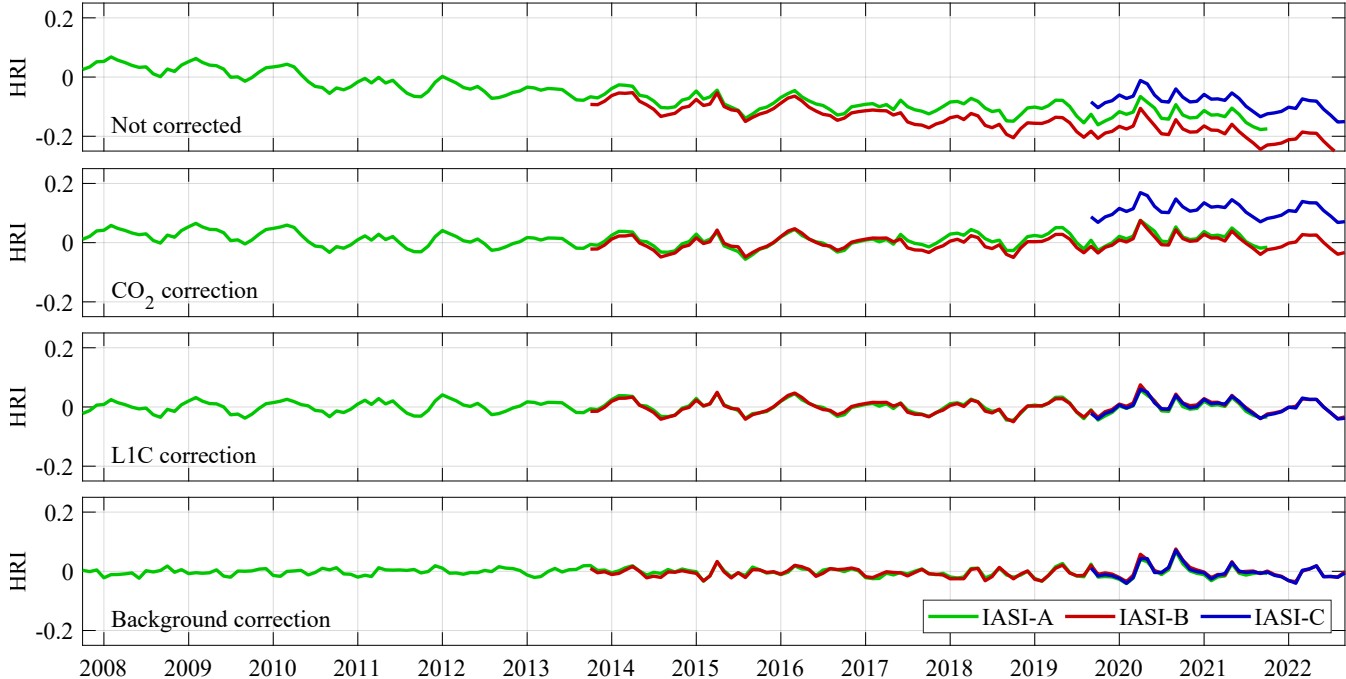

**Figure 5.** Monthly average HRI time series over 10 remote regions for the three IASI instruments separately. The top panel shows the uncorrected timeseries, and the other panels, from top to bottom, show the effects of the corrections that are applied consecutively.

cloud product, it inherits all its advantages as a proven and well-validated product. The NN utilises carefully selected IASI

channels as input (excluding channels affected by long-lived tracers $CO_2$, $N_2O$, $CH_4$, CFC-11 and CFC-12) and was shown to be temporally consistent, and coherent across the three IASI instruments. The network presented in Whitburn et al. (2022) was trained to distinguish completely clear scenes (0% cloud cover) from the rest. For the $NH_3$ processing, two additional networks were trained to distinguish scenes with a cloud cover below 10% and 25% respectively. With this, three cloud flags are available and these have now been integrated in the v4 of the reanalysed ANNI-$NH_3$ product. The results presented in the

rest of this paper utilise the flag corresponding to the 10% threshold.

### 4.4 Residual bias corrections

The stability of the HRI was evaluated over ten remote regions, where only background columns of $NH_3$ are expected. Their average monthly HRI is shown in Fig. 5 for the three IASI instruments separately. The top panel shows the average as obtained with the HRI setup as described above, i.e. with generalized inverse and with a $CO_2$ Jacobian. As with previous versions of

the product, a spurious linear trend is observed, but thanks to the introduction of the $CO_2$ Jacobian, its magnitude is reduced to about 0.01 per year, compared to 0.03 per year previously. A slightly steeper decrease is observed for Metop B. We correct for these trends by adding a time-dependent offset as in Van Damme et al. (2021). The result after correction is shown in the second panel of Fig. 5.





A detailed analysis was made of this time series to detect offsets between the different instruments and shifts that coincide

with known changes in the IASI L1C data. The largest of these is the offset of 0.11 seen between IASI-C and the two other instruments. Small offsets in the HRI time series of IASI-A were found in 2010, 2015 and 2017 and in the HRI of IASI-B in 2015. For each of these, offset corrections were calculated in the range of 0.01–0.03. Thanks to the generalized inverse, their magnitude is drastically reduced (previously, offsets as large as 0.6 were observed). The resulting corrected time series is shown in the third panel. This time series is temporally stable and shows an excellent consistency between the three instruments,

but exhibits a weak seasonal cycle, likely due to the combined effect of seasonal changes in the concentrations of $H_2O$ and volatile organic compounds that absorb in the same spectral range as $NH_3$. To remove this seasonality an offset depending on latitude and month of the year was calculated from 2012–2014 IASI-A data and applied on all data. The HRI after correction is shown in the bottom panel of Fig. 5. Thanks to the improved setup of the HRI, and the new cloud product, the magnitude and therefore also uncertainty of all these corrections is lower than in the previous product, which in the end results in much

improved temporal consistency.

## 5    Other changes to the retrieval network

An additional change in the setup of the HRI concerns the choice of the spectra used for determining the mean background spectra and its associated covariance matrix. As before (Franco et al., 2018) we use a random selection of IASI spectra from the year 2013, but now with a proportionally larger number of spectra from selected parts of the Saharan, Arabian, Great Australian

and Namib desert. It was found that this was an efficient way for countering the small negative biases that are seen over these areas and that are associated with surface emissivity variations. It also leads to a better detection of $NH_3$ transport over deserts.

Since ANNI v2 (Van Damme et al., 2017) the reanalysis product relies on a surface temperature retrieved from a custom-built neural network, rather than the IASI L2 surface temperature. With ANNI v4, this network has been retrained from data that were generated using latest version (v6.5) of the IASI L2 algorithm. The input parameters of the NN for the retrieval of surface

temperature include 60 selected baseline channels (a subset of the channels used in the cloud NN), surface altitude, total water vapour column and the three output values of the cloud NNs. Mean and standard deviation of the difference between the L2 surface temperature and that retrieved from the network are of the order of 0.5 K and 1.5 K for cloud fraction up to 25%.

A final series of changes concern the network architecture and training database. In previous versions, separate neural networks were employed for the retrieval over land and ocean. These were trained respectively with Gaussian a priori profiles

peaking at the surface, and a larger, more general one, with profiles peaking at different altitudes. However, a careful comparison showed that the more general network performed as good as the network trained specifically for profiles peaking at the surface. For this reason, only one network was trained for version 4, for a priori profiles peaking at altitudes $z_0$ from 0 to 20 km, with a width $\sigma$ in the range of 0.1 to 3 km. In view of the averaging kernel calculation, 20% of the profiles of the training database have an $NH_3$ profile with a $\sigma$ of 100 m.

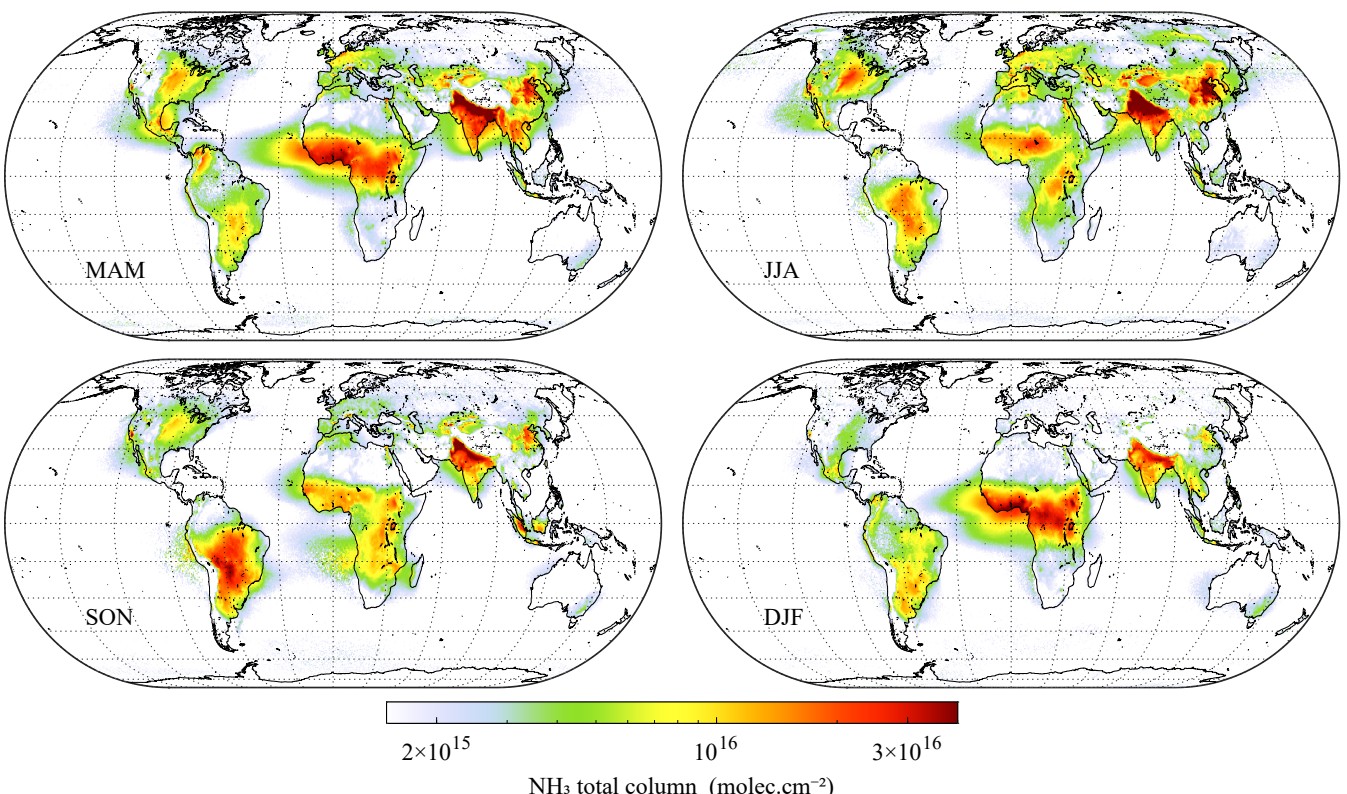

**Figure 6.** NH$_3$ seasonal average, derived from $0.5° \times 0.5°$ monthly averages of the reanalysis product of ANNI v4. Data includes all measurements from IASI-A (October 2007 to December 2019), IASI-B (March 2013 to September 2022) and IASI-C (September 2019 to September 2022), with a cloud fraction below 10 %.

## 6  Evaluation

### 6.1  Comparison with version 3

As outlined before, v4 has an improved temporal consistency compared with v3. In this section, we provide a short assessment of the new NH$_3$ spatial distributions and how they compare with previous versions. As an illustration of the new product, a seasonal average over 2007–2022 is presented in Fig. 6. The distributions follow closely the ones of previous versions (Van Damme et al., 2015). Comparisons with version 3 are provided in Fig. 7 and Fig. 8. The main differences are: (1) Overall larger columns, especially in areas with high columns. As explained in Sect. 4.1, this is due to the more robust way of calculating the HRI, which makes it less sensitive to small errors in the forward model. Comparing individual observations, with an HRI above 3, the new version is about 15% to 20% larger. (2) A notable improvement over high latitudes thanks to the improved bias correction (see Sect. 4.4). Averaged columns were clearly overestimated in ANNI v3 for such observations, especially over Greenland and Antarctica, but also Canada and Russia. (3) Slightly higher concentrations over deserts, in part due to the





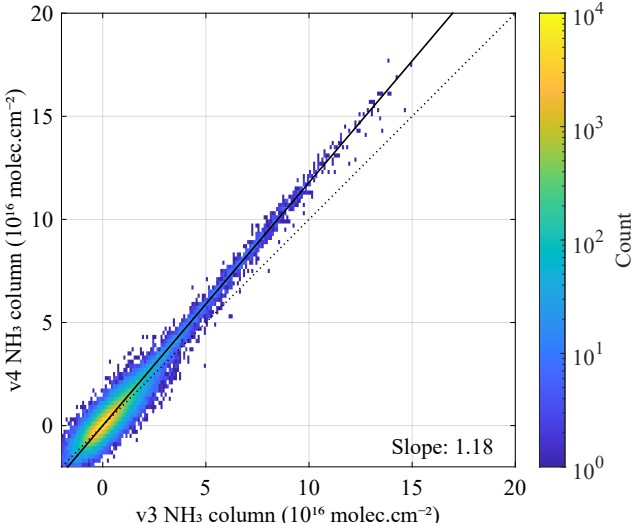

**Figure 7.** Comparison between retrieved columns with ANNI v3 and v4 for all morning observations of 17 June 2015. The slope was determined from observations with an HRI above 3.

overall increase in v4, in part due to the larger weight of deserts in the construction of mean background spectra and associated covariance matrix (see Sect. 5). In v3, negative HRIs were consistently observed over certain deserts, resulting in negative average columns, or low biases. This problem is not entirely gone (e.g. the average HRI over parts of the Arabian desert is still negative), but much improved, resulting in more consistent pronounced transport patterns over e.g. the Saharan desert. (5)

Overall larger concentrations over oceans. The improved bias correction (last step of the HRI correction presented in Sect. 4.4) enables to remove practically all negative values on a long term average. Average columns over remote ocean are now around $7 \cdot 10^{14}$ molec.cm$^{-1}$ almost everywhere. An adverse effect of the correction might be overestimated columns over the Red Sea, Persian gulf and Mediterranean Sea.

The most obvious remaining artefact in the v4 distribution concerns the continuity of the land-sea transitions. While they are

reasonable for some regions of outflow (Gulf of Mexico, Mediterranean Sea), off the West coast of Africa, over the Arabian Sea, Gulf of Bengal or Yellow Sea, the transition is too abrupt to be realistic. The origin of this problem is that different $NH_3$ profiles are used for land and ocean. With the introduction of the AVKs, this does not constitute a problem in model comparison or assimilation. However, for stand-alone use of the product, it would be desirable in the future to improve the parametrization of the a priori vertical profile shape.

## 6.2 Comparison with an optimal estimation retrieval

Given the low bias in ANNI $NH_3$ v3, it is important to exclude the presence of other biases related to the HRI calculation. Here, we present the results of an independent intercomparison that was conducted between the ANNI v4 retrieval output and that of an optimal estimation approach which relies on spectral fitting.







**Figure 8.** Comparison between the NH$_3$ columns of the near real time products of ANNI v3 (top) and v4 (bottom) on a 0.5° × 0.5° grid. Data includes all morning IASI-A data from 2008–2018, with a cloud fraction below 10 %.





**Figure 9.** Comparison between ANNI v4 $NH_3$ columns and retrievals based on optimal estimation for two scenes, one over Europe (top panels, 18 April 2013, Metop A morning overpass) and North America (bottom panels, 6 May 2021, Metop B morning overpass). The left panels depict the optimal estimation retrieved columns. The middle panels are scatter plots between the two retrievals, where each observation is colour coded according to thermal contrast (brightness temperature of the surface minus the temperature at half the boundary layer height). The right panels summarize the comparison by means of histograms of the differences.

For the optimal estimation retrieval, the Atmosphit forward and inverse model was used (Coheur et al., 2005), which is the same tool whose forward model is used for the construction of the ANNI training database. The optimal estimation was set




up as follows. The retrieval range was set to 900 to 975 $cm^{-1}$. Total columns of $NH_3$ were retrieved with a fixed vertical profile, using the same parametrization as in ANNI NRT. The $NH_3$ variance was set to 1000 %, corresponding to an almost unconstrained retrieval. Together with $NH_3$, $H_2O$ was retrieved in 10 partial columns, with the a priori coming from the IASI L2. Total columns of $CO_2$, $O_3$ and CFC-12 were retrieved as well as the surface temperature. Spectral emissivity was taken

from Zhou et al. (2013). Before presenting the results, it should be emphasized that despite the similarities in both retrieval approaches (same input parameters, vertical profiles, forward model), no perfect agreement is expected because of: (1) use of a narrower spectral range in the optimal estimation retrieval; (2) different propagation of instrumental noise to the retrieval result; (3) limitations of the fitting model (e.g. with respect to fitting water vapour or surface emissivity); (4) errors related to the imperfect training of the neural network.

For the comparison, two days were selected, one over Europe and one over North America, with relative high $NH_3$ columns. The results are shown in Fig. 9. Intercepts, mean and median differences are all of the order of $10^{15}$ molec.$cm^{-1}$ or below. Regression slopes, calculated using iteratively reweighted least squares to remove the impact of outliers, are 0.97 and 1.05. While the scatter around the 1–1 lines is not negligible (with standard deviation of the differences around 3–4 $\cdot 10^{15}$ molec.$cm^{-1}$), these numbers demonstrate the overall consistency of both retrieval approaches and do not indicate a significant bias.

## 445  7  Uncertainties

### 7.1  Propagation of uncertainty

In previous ANNI versions, an estimated uncertainty $\sigma_{\hat{X}}$ was calculated for each individual measurement $\hat{X}$ via (Ku, 1966)

$$\sigma_{\hat{X}}^2 = \sum_i \left( \frac{\partial \hat{X}}{\partial p_i} \right)^2 \sigma_{p_i}^2, \tag{61}$$

with $\sigma_{p_i}$ the uncertainties of the different input parameters $p_i$. This formula assumes uncorrelated uncertainties, but as this

cannot always be justified, in ANNI v4, we switch to the more general (Tellinghuisen, 2001)

$$\sigma_{\hat{X}}^2 = \boldsymbol{J}^{\mathrm{T}} \mathbf{S}_p \boldsymbol{J}, \tag{62}$$

with $\mathbf{S}_p$ the error covariance matrix of the input parameters (with covariances $S_{p,ij} = \sigma_{p_i p_j}$) and $\boldsymbol{J}$ the Jacobian of the retrieval, with components $\frac{\partial \hat{X}}{\partial p_i}$.

    In the ANNI retrieval framework, the input parameters include the skin temperature, the surface pressure, the HRI, the

surface emissivity, the zenith angle, the width and the peak of the Gaussian vertical $NH_3$ profile, the temperature profile (15 levels) and the water vapour profile (7 levels). After some preliminary analysis, it was concluded that only the correlations between the uncertainties in the temperature profile cannot be neglected. We therefore employ a block diagonal covariance matrix, block diagonal for the elements pertaining to the temperature profile, and diagonal for all other input parameters. As for uncertainty on the vertical profile, this source of uncertainty is removed when applying averaging kernels. For this reason,

uncertainties are reported with and without the vertical profile uncertainty, to be used according to whether or not AVKs are applied.





## 7.2 Random and systematic uncertainties

In total, we report four types of uncertainty for each observation: random or systematic, and with or without the vertical profile uncertainty included. Reporting random and systematic uncertainties separately, is a generally recommended practice (Boersma et al., 2004; Merchant et al., 2017; Sayer et al., 2020). Random uncertainties describe errors specific to a single measurement, and assuming a normal distribution, these average out over many repeated measurements. Systematic uncertainties are those that exhibit correlations in time or space, and are thus associated with more than one measurement. This type of errors can lead to biases in the measurement dataset. In ANNI v4, both random $\sigma_{r\hat{X}}$ and systematic $\sigma_{s\hat{X}}$ uncertainties are calculated using Eq. (62) and estimates of the random and systematic uncertainties/covariances of the input parameters.

Random and systematic uncertainties can be combined and averaged in different ways, according to the needs of the user. In particular, for a given measurement $\hat{X}$, a total uncertainty estimate can be obtained as (Gomez-Pelaez et al., 2013)

$$\sigma_{\hat{X}}^2 = \sigma_{r\hat{X}}^2 + \sigma_{s\hat{X}}^2. \tag{63}$$

An average measurement uncertainty can be associated with an average $\bar{X}$ of a series of $n$ measurements $\hat{X}_i$ as

$$\sigma_{\bar{X}}^2 = \sigma_{r\bar{X}}^2 + \sigma_{s\bar{X}}^2 \tag{64}$$

$$= \frac{1}{n^2} \sum_{i=1}^n \sigma_{r\hat{X}_i}^2 + \frac{1}{n} \sum_{i=1}^n \sigma_{s\hat{X}_i}^2. \tag{65}$$

Note here especially the difference between $n^2$ and $n$ in the denominators. For the special case where all random and systematic uncertainties are the same, we obtain

$$\sigma_{\bar{X}}^2 = \frac{1}{n}\sigma_{r\bar{X}}^2 + \sigma_{s\bar{X}}^2, \tag{66}$$

which tends to the expected $\sigma_{\bar{X}} = \sigma_{s\bar{X}}$ for large $n$.

## 7.3 Uncertainties of the input parameters

As most input parameters come without uncertainty budget, let alone covariances, we made best-effort estimates of the co(variance) based on the limited information that is available. For now, the same (co)variances were used for the near-real time as for the reanalysed $NH_3$ product. It is also important to note that the systematic uncertainties of the input parameters vary according to the time and space scales that are considered (Boersma et al., 2004; Merchant et al., 2017; Sayer et al., 2020). Temperature profiles for example, may be more biased monthly than annually. Here, we estimate systematic uncertainties with a typical L3 gridded data product in mind, i.e. for spatial scales of the order of one degree latitude/longitude or less, and for time periods of the order of one month or less.

The (co)variances, summarized in Table 2, were determined as follows:

**HRI** By definition, the random uncertainty on the HRI equals one. We estimate a systematic uncertainty of $0.1$ due to potential and residual interferences (e.g. surface emissivity, VOCs). To account for potential biases in the spectroscopy and generalized error covariance matrix, we add to this an additional $10\%$ on the calculated HRI value.





**Table 2.** Estimated random and systematic uncertainties of the input parameters.

| Component | Random $\sigma_r$ | Systematic $\sigma_s$ |
|---|---|---|
| HRI | 1 | $0.1 + 10\%$ |
| Surface temperature (K) | 1.5 | 0.5 |
| Emissivity | 0.01 | 0.005 |
| Temperature profile, land (K) | $1 - 2$ | $0.5 - 1$ |
| Temperature profile, sea (K) | $0.5 - 1$ | $0.5 - 1$ |
| Surface pressure (Pa) | 500 Pa | 250 Pa |
| Water vapour profile | $10 - 20\%$ | $5 - 10\%$ |
| $NH_3$ profile peak altitude (m) | 200 | 100 |
| $NH_3$ profile width (m) | 200 | 100 |

**Skin temperature** Random and systematic uncertainties were set to 1.5 and 0.5 K respectively. These values are in line with the difference between the IASI L2 skin temperature product and the dedicated neural network used for the reanalysis product of ANNI (see Sect. 5).

**Emissivity** For emissivity, which originates from the monthly climatology of Zhou et al. (2013), an uncertainty of 0.01 and 0.005 was assumed for respectively the random and systematic components.

**Temperature profile** Variances were set based on validation results of the IASI level 2 (Eumetsat, 2021): systematic uncertainties of 1 K for the surface level and 0.5 K for the other levels; random uncertainties of 2 K for the surface level and 1 K for the other levels for land observations and 1 K for the surface level and 0.5 K for the other levels for ocean obser-
vations. Covariance matrices were then built by appropriate scaling of correlation matrices. These were built, based on a statistical analysis of the differences between collocated ERA5 and IASI L2 profiles. Correlation coefficients were set to 0.5 between neighbouring levels, and 0.25 between levels that are two levels apart. Above 10 km, no strong correlations were observed, and the covariance was therefore assumed to be diagonal for these levels.

**Water vapour profiles** Relying again on the IASI level 2 validation report (Eumetsat, 2021), random uncertainties were set
to 10 % below 3 km and 20% above. Systematic uncertainties were set to half these numbers.

**Surface pressure** A random and systematic uncertainty of 500 and 250 Pa was used.

**$NH_3$ profiles** The uncertainties related to $NH_3$ profile are characterized by uncertainties on the width and the peak of the Gaussian shaped vertical profile. Random and systematic uncertainties of 200 and 100 m were used for both parameters. Given the short lifetime of $NH_3$ in the atmosphere, these are likely of the right order of magnitude. To obtain better
estimates in the future, a thorough analysis using in situ measurements or modelled profiles would be desirable.





### 7.4 Uncertainty budget of $NH_3$

It is useful, remembering the general form $\hat{X}^a = \mathrm{HRI}/\mathrm{SF}^a + B$ of the retrieval, to rewrite the propagation of uncertainty in terms of the uncertainty of the nominator and denominator (see also Boersma et al. (2004); van Geffen et al. (2022)). Neglecting the small dependence of the SF on the HRI, we obtain


$$\sigma_{\hat{X}}^2 = \left(\frac{\partial \hat{X}}{\partial \mathrm{HRI}}\right)^2 \sigma_{\mathrm{HRI}}^2 + \left(\frac{\partial \hat{X}}{\partial \mathrm{SF}}\right)^2 \sigma_{\mathrm{SF}}^2 \tag{67}$$

$$= \frac{\sigma_{\mathrm{HRI}}^2}{\mathrm{SF}^2} + \frac{\sigma_{\mathrm{SF}}^2}{\mathrm{SF}^2}(X - B)^2. \tag{68}$$

Taking into account both random and systematic uncertainties, we see from Table 2 that the uncertainty on the HRI has an absolute (constant) and a relative (proportional to the value of the HRI) component, so that

$$\sigma_{\hat{X}}^2 = \frac{\sigma_{\mathrm{abs,HRI}}^2}{\mathrm{SF}^2} + \frac{\sigma_{\mathrm{rel,HRI}}^2}{\mathrm{SF}^2} + \frac{\sigma_{\mathrm{SF}}^2}{\mathrm{SF}^2}(X - B)^2 \tag{69}$$


$$= \underbrace{\frac{1^2 + 0.1^2}{\mathrm{SF}^2}}_{\sigma_{\mathrm{abs}}^2} + \underbrace{\left(0.1^2 + \frac{\sigma_{\mathrm{SF}}^2}{\mathrm{SF}^2}\right)(X - B)^2}_{\sigma_{\mathrm{rel}}^2}. \tag{70}$$

#### 7.4.1 Absolute uncertainty contribution

The first term is in the optically thin limit independent of the HRI and thus the column, and solely depends on the scene conditions:

$$\sigma_{\mathrm{abs}} = \frac{\sigma_{\mathrm{abs,HRI}}}{|\mathrm{SF}|} = \frac{\sqrt{1 + 0.1^2}}{|\mathrm{SF}|} \approx \frac{1}{|\mathrm{SF}|}. \tag{71}$$

It is this term that is used as part of the post-filter to determine whether there is enough intrinsic sensitivity (thermal contrast) to make a valid measurement, i.e. one whose uncertainty is not completely overwhelmed by the instrumental noise. Currently, the post-filter threshold is set to $\sigma_{\mathrm{abs}} < 1.5 \cdot 10^{16}$ molec.cm$^{-2}$. Note also that a scene-dependent detection threshold of the measurements (typically taken as HRI>3), is conveniently expressed in terms of the absolute uncertainty as $X_{\mathrm{thres}} = 3\sigma_{\mathrm{abs}}$.

The absolute uncertainty contribution is illustrated on the left panels of Fig. 10 for the IASI morning overpass (land observa-
tions between 60° S and 60° N), as a function of thermal contrast (TC). As before, we define TC as the brightness temperature of the surface minus the temperature at half the boundary layer height. The absolute uncertainty starts from around $1 \cdot 10^{15}$ molec.cm$^{-2}$ and increases as expected with decreasing thermal contrast, with a global median of $4 \cdot 10^{15}$ molec.cm$^{-1}$. Observing the inverse proportionality with thermal contrast, the following empirical formula can be used to obtain ballpark estimates of the absolute uncertainty or sensitivity of the IASI $NH_3$ retrieval (for positive thermal contrasts):


$$\sigma_{\mathrm{abs}} = \frac{3.6 \cdot 10^{16}}{\mathrm{TC}} \frac{\mathrm{molec. \, K}}{\mathrm{cm}^2} \tag{72}$$





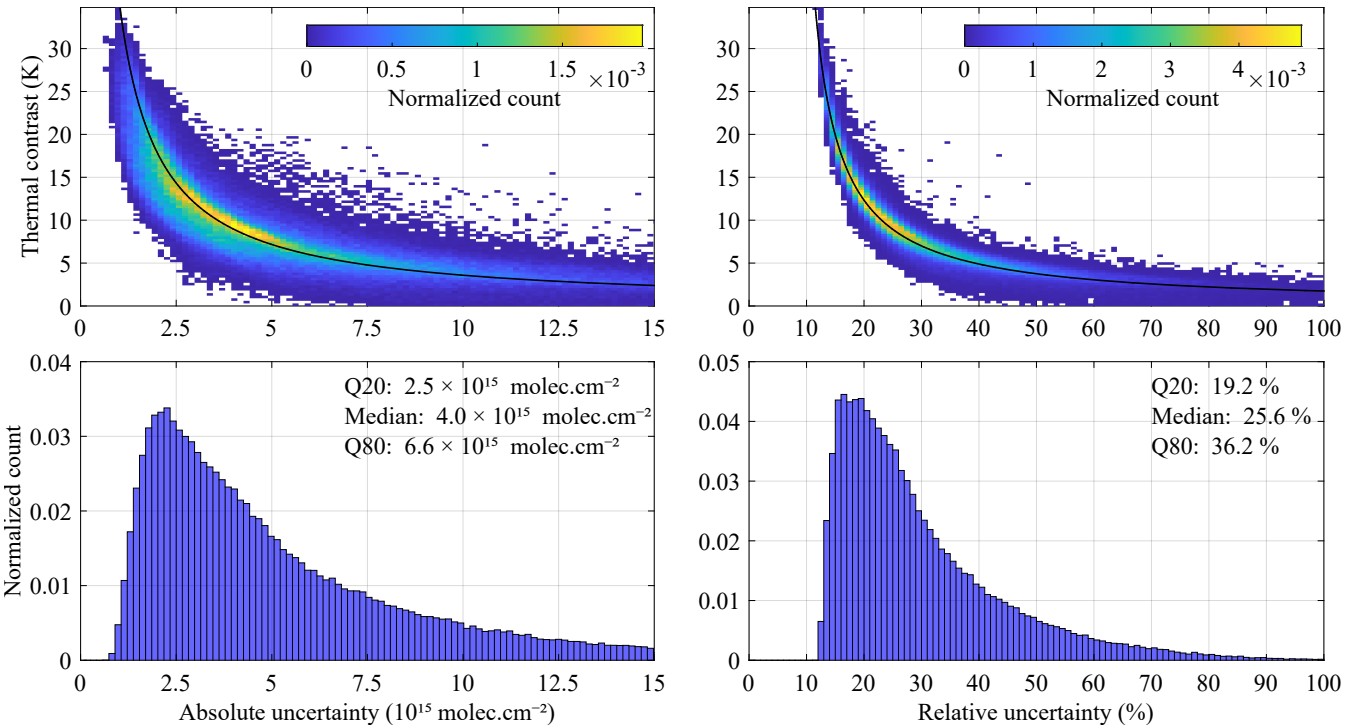

**Figure 10.** Illustration of the absolute (left) and relative (right) components of the retrieval uncertainty. The top panels illustrate their dependence on thermal contrast, the bottom panels show the normalized count. Data in this plot originates from IASI-B observations on 15 January, April, July and October 2021, morning overpass, land only and between 60° S and 60° N. The approximations from Eq. (72) and Eq. (76) are in shown in black in the top panels.

The constants were determined from a fit of the data shown in Fig. 10. Expressed in terms of Q20 and Q80 quantiles the estimated absolute retrieval uncertainty of IASI (mid-latitude, land, morning overpass) can also be summarized as

$$\sigma_{\text{abs}} = [2.5 - 6.6] \cdot 10^{15} \text{molec.cm}^{-2}. \tag{73}$$

### 7.4.2 Relative uncertainty contribution

The term $\sigma_{\text{rel}}$ is proportional to the column and hence expresses a relative uncertainty for fixed atmospheric conditions:

$$\sigma_{\text{rel}} = \sqrt{0.1^2 + \frac{\sigma_{\text{SF}}^2}{\text{SF}^2}}(X - B) \approx \frac{\sigma_{\text{SF}}}{|\text{SF}|}(X - B), \tag{74}$$

or

$$\frac{\sigma_{\text{rel}}}{(X - B)} = \frac{\sigma_{\text{SF}}}{|\text{SF}|}. \tag{75}$$


This term is illustrated on the right panels of Fig. 10. Again, we observe an inverse proportionality with thermal contrast, which can be approximated as

$$\sigma_{\text{rel}} = \left( 0.07 + \frac{1.6\text{K}}{\text{TC}} \right) \text{NH}_3.$$ (76)

For typical morning land observations, the relative contribution to the uncertainty starts from around $14\%$ (corresponding to a TC of 20 K). Expressed in terms of Q20 and Q80 quantiles, the estimated relative retrieval uncertainty of ANNI (mid-latitude, land, morning overpass) can be summarized as

$$\sigma_{\text{rel}} = [19 - 36]\%\text{NH}_3.$$ (77)

## 8 Conclusions

In this paper, we presented v4 of the $\text{NH}_3$ ANNI retrieval. The most important change is the introduction of averaging kernels, which will greatly ease future model assimilation and comparisons with independent measurements or model output. Most other changes to ANNI v4 contribute to the overall consistency of the product. An example is the incorporation of the temporally consistent cloud flag. The improved way of calculating the HRI makes the product more robust across the different IASI instruments and more temporally harmonious. Importantly, the HRI became also less sensitive to small errors in the forward model related to the instrumental line shape function. Previous versions were biased low by some 15–20 % due to such errors. Theoretically we can now exclude the existence of large biases of this sort. We also demonstrate this with an optimal estimation experiment. In addition to the AVKs, we revised the uncertainty calculation and now provide better and more comprehensive information on the expected error of the measurement. We also show how the retrieval uncertainty contains a part proportional to the column and a part that is independent of the column. In the near future, the most important changes will gradually be implemented for all the other tracers retrieved with ANNI (AVKs, the use of generalized covariance matrices and the better treatment of uncertainties).

*Data availability.* The IASI-$\text{NH}_3$ datasets are available from the Aeris data infrastructure (http://iasi.aeris-data.fr/NH3).

*Author contributions.* L.C. led the research, conceptualized the ANNI retrieval changes and wrote the first version of the manuscript. B.F., L.C., J.H.-L., D.H., S.W., M.V.D. contributed to the code or data processing. L.C., M.V.D., T.D.G. and L.N. prepared the figures. T.D.G., M.V.D. and L.C. implemented the corrections presented in Sect. 4.4. L.C., M.V.D. and B.F. developed the improved treatment of uncertainties. All authors took part in discussions and revised the manuscript.

*Competing interests.* No competing interests are present.



570 *Acknowledgements.* The research was co-funded by the Belgian State Federal Office for Scientific, Technical and Cultural Affairs (Prodex HIRS), the Air Liquide Foundation (TAPIR project), EUMETSAT (AC-SAF project) and ESA (Short-lived greenhouse gases CCI project). This work is also partly supported by the FED-tWIN project ARENBERG ("Assessing the Reactive Nitrogen Budget and Emissions at Regional and Global Scales") funded via the Belgian Science Policy Office (BELSPO). L.C. is Research Associate supported by the Belgian F.R.S.-FNRS. L.C. is grateful to Tim Hultberg for pointing out the necessity of using a generalized inverse for the calculation of the HRI.



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
