# Peer review of "The IASI NH3 version 4 product: averaging kernels and improved consistency"

_Atmospheric Measurement Techniques, 2023_

## Author Comment (AC1)

We are very grateful to the reviewer for the positive assessment of the manuscript, valuable comments and helpful suggestions. All comments have, where possible, been addressed or answered. Replies are inline in blue.

**RC1: 'Comment on amt-2023-48', Anonymous Referee #1, 26 Jun 2023**

The authors have presented version 4 of the IASI NH3 product. In this version, they introduced total columns averaging kernels. With the averaging kernels, the IASI NH3 can be easier to be compared with model simulations and can be better implemented for the NH3 assimilations in the models. The improvement of HRI calculations makes the product more robust across the different IASI instruments. The manuscript is clearly written and well-structured and will be of interest to scientists using the IASI NH3 observations. I therefore recommend publishing this manuscript in AMT after addressing some minor comments below.

Minor comments:

L55: eq (1) and (2) are used to calculate the HRI and estimate true NH3 total column. According to the context, I guess that mean spectrum includes the NH3 background column B. Are they related? Maybe a short sentence to describe the relation between mean L and background B?

This is correct. A short sentence has now been added to clarify that the background B corresponds to the mean NH3 in mean L.

L75-77: eq (4). Can you explain a little bit more on why you introduce the filter? After applying the filter, all low values will be removed and when averaging the product over a period, this will introduce a positive bias, which will affect a trend analysis of the NH3.

In fact, the filter does not preferentially remove low values: it removes observations for which the scaling factor is smaller than a certain threshold, irrespective of the amount of NH$_3$ (as the scaling factor is the ratio between HRI and column). Small scaling factors correspond to situations where there is little or no measurement sensitivity.

L164: revise ' take care off' to 'take care of'

This has been corrected.

L281 you have mentioned 'N' here. This is the first time you introduced 'N'. I guess this is the normalization factor. Please specify it. Figure 3 shows the AVK normalization factors. Please mention that this is the 'N' variable in the caption.

This has been added and clarified (also in the caption).

L395 what do you mean by a larger here? Can you give a clearer explanation here?

We meant "more general" instead of larger. It is more general, as the network is trained for a larger set of vertical profiles. We removed the word "larger" now.

Figure 8. Can you also add a sub-figure to show the difference of the two versions? It will be easier to see the distribution of the difference.

A subfigure has now been added with the difference.

Section 6.2 presented the intercomparison between ANNI v4 and results of an optimal estimation approach. Authors shows the consistency of both retrieval approaches. After reading the section, I am wondering why use the NN method instead of the optimal estimation approach? Maybe add some discussion about it.

The NN method approach has numerous advantages over the optimal estimation approach (the most straightforward one is computational efficiency, but there are several others). These are discussed and summarized in section 7 in Whitburn et al. (2016). As section 6.2 in the current paper is mainly meant to exclude the existence of biases, we did not want to discuss (dis)advantages of either approach here. However, in view of your comment, we have now added a sentence referring to Whitburn et al. (2016).

---

## Author Comment (AC2)

We are very grateful to the reviewer for his positive assessment of the manuscript, valuable comments and helpful suggestions. All comments have, where possible, been addressed or answered. Replies are inline in blue.

**RC2: 'Comment on amt-2023-48', Daven Henze, 14 Jul 2023**

The manuscript by Clarisse et al. describes a new NH3 retrieval from IASI, which has the improvement of including averaging kernels as well as fixing some other consistency issues. The introduction of averaging kernels for the standard IASI product is a welcome addition to the retrieval product that has been on the wishlist of potential users for more than a decade, as it allows for models to be fairly compared to the satellite retrieval data. This is thus an important and significant step forward and will likely ensure that IASI NH3 data is used in more quantitative model-based studies in the future. Overall, the article is very well written and logically sound. The recapitulation of some theory is highly appreciated, as it provides a self-contained, consistent overview of DOAS and optimal estimation retrieval theory and notation. My main request is that the article touch on, if only cursorily, how the updates made here are expected to impact comparisons of IASI data to in situ measurements (where not directly comparable, one can consider spatial gradients, season and annual trends, etc.). Following the addition of such content, and addressing a few small issues below, the article is suitable for publication in AMT.

Major comment:

While it is interesting to compare to the v3 retrieval and an optimal estimation retrieval, there's no comment on how the updates may address any comparison to in situ measurements. I recognize that column NH3 observations aren't available from many in situ measurements, but ground-based surface concentration measurements can be used to evaluate gradients, seasons trends, and annual trends. Models can also be used as a transfer function to relate in situ measurements to satellite-observed column concentrations, as a means of evaluation. While a detailed comparison using models may be beyond the scope of a retrieval algorithm paper, some discussion of bias and error relative to in situ measurement is still warranted, and feasible. Thus I'd request the following questions be addressed at a minimum: based on all prior application studies with the IASA products, was there a suspected under-estimation of NH3, especially in more remote areas? Is v4 an improvement in this regard? To what extent does the new temporal trend correction improve comparison of trends to in situ measurements?

We agree that these are important questions, but we think that a comprehensive discussion or validation is well outside of the scope of the present paper. We have now added a paragraph discussing validation and possible underestimation of the IASI NH$_3$ product. However, we did not touch on the complex topic of trend comparisons of satellite columns vs in situ measurements, as, from our experience, these are very difficult to make, given the limited spatial representativity of in situ measurements and the fact that NH$_3$ columns are much more impacted by long term trends of other inorganic pollutants (NO$_2$, SO$_2$) than in situ concentrations (e.g. Lachatre et al., 2019). The paragraph that was added reads:

*The last detailed global validation of the IASI NH$_3$ product was based on a comparison of ground-based FTIR measurements of NH$_3$ with the LUT-based NH$_3$ product, where a low bias around 35% was found (Dammers et al., 2016). Since then, two independent validation studies have been conducted. One study (Guo et al., 2021) compared IASI ANNI v3 with in situ measurements in Colorado, U.S. and found regression slopes ranging from 0.78 to 1.1, and intercepts of the order of $1\cdot10^{15}$ to $2\cdot10^{15}$ molec./cm². The second study (Wang et al., 2022) compared IASI NH$_3$ columns with columns obtained from FTIR measurements in Hefei, China. Here, mean differences around $3.5\cdot10^{15}$ molec./cm² (IASI being lower)*

*were found and regression slopes close to one. Given the results of the comparison with the OEM method, we do not expect any significant bias in v4 for columns above $1 \cdot 10^{16}$ molec./cm² in comparisons that correct for the vertical profile assumption of the retrieval. A comprehensive validation of the v4 product is foreseen within the framework of ESA's aerosol and ozone precursor project, that should confirm this, as well as assess the performance of the algorithm on low columns.*

Reference: M. Lachatre et al. (2019) "The unintended consequence of $SO_2$ and $NO_2$ regulations over China: increase of ammonia levels and impact on $PM_{2.5}$ concentrations," Atmos. Chem. Phys., vol. 19, no. 10, pp. 6701–6716, 2019, doi: 10.5194/acp-19-6701-2019.

Minor comments and edits:

61: What is the reason for applying a scaling factor as 1/SF * HRI rather than SF * HRI? Is there a numerical benefit to this formulation?

In earlier versions the SF was defined the other way around. It was changed in v2, for reasons related to the training of the network. We refer to Van Damme et al. (2017) for a detailed explanation. In short: when COL=HRI/SF, large SF correspond to favorable measurement conditions; SF near zero to limited measurement sensitivity. During training, the cost function of the neural network is minimized with respect to difference in scaling factor. With the old definition, measurements with no sensitivity had a very large SF (~1/zero) and could dominate the training process. With the current definition of SF, measurements with the best sensitivity carry the most weight in the training, while measurements with no sensitivity least.

75-80: What is the origin / significance of the factors of 1.5 in the filtering criteria Eq. (4) and (5)? Also, is my reading of the Eq. (5) filter correct in that negative column concentrations are allowed but only when HRI is not more than 50% above background? Lastly, what is the background value, B? Is it actually zero or an arbitrarily small number?

Both constants were set by looking at both single and averaged data, to exclude unphysical data. That is: the negatives and very large values over remote areas should average out, but if they are so large that they do not, the postfilter should remove them. As for Eq. (5), this puts a cap specifically on negative columns. In general, negatives occur when the sign of the HRI is opposite that of the thermal contrast (e.g., positive TC and negative HRI) and there are very good reasons for conserving these negatives (see Whitburn et al., 2016 and Clarisse et al., 2019). Negatives are mostly due to noise on the HRI (resulting in an HRI of the "wrong" sign) and such negatives average out over many measurements. However, they can also be caused by errors in the assumed vertical $NH_3$ profile or errors in the temperature profile/surface temperature, in which case they do not average out. Eq. (5) specifically targets such cases by excluding observations with large HRIs and negative columns. For the final question: the background B is 0 for v4, and we have now made this clearer in the manuscript. In future version it could be set to a small value $\sim 10^{14} - 10^{15}$ molec.cm$^{-2}$ to reflect columns found in the most remote areas. We have added a sentence in the manuscript referring to see Whitburn et al., 2016 and Clarisse et al., 2019 for a discussion on negative columns.

Ref. L. Clarisse et al., "A Decadal Data Set of Global Atmospheric Dust Retrieved From IASI Satellite Measurements," J. Geophys. Res. Atmos., vol. 124, no. 3, pp. 1618–1647, Feb. 2019, doi: 10.1029/2018jd029701.

164: off —> of

This has been corrected.

As a user of DOAS and optimal estimation retrievals, I found Sections 3.3 and 3.4 very valuable, in terms of drawing similarities and differences between the two types of averaging kernels and how they are used. The additional discussion of the two approaches for making "apples to apples" comparisons of modeled to retrieved values (and the note on when they are equivalent, as a ratio) is, while not new, extremely useful for new users to have laid out in a single paper with consistent and clear notation.

Thank you very much for this comment.

287 and Fig 3: I agree the deviations of the overall set of samples from unity is small. It would though be interesting to see if there are regional biases. Showing only a single day makes it hard to discern any gradients between the missing values, and also for this day in particular many global hotspots are omitted. Could the authors instead or additionally show means for normalization factors averaged across a much longer period?

We have now added a panel with a year average. The results confirm what was observed on a single day. Small biases are observed though at high latitude and in regions affected by low clouds. We revised the discussion of this figure in view of the update.

Section 4.1: I think the authors means pseudo-inverse, not generalized inverse. A generalized inverse is a more general, less restrictive, non-unique notion of matrix inverse. I don't think that's what they are using here. Rather, I note the way they are using a truncated eigenvalue decomposition to represent the inverse is the same as a truncated SVD decomposition (given that S is symmetric, and the eigenvalues position), which is based on the pseudo-inverse that omits all vectors associated with zero (or effectively zero, numerically) singular values. I also note that Rodgers refers to this as a pseudo-inverse. I'm not familiar with their other reference.

Thank you for this comment. The pseudoinverse is a special case of a generalized inverse (https://en.wikipedia.org/wiki/Generalized_inverse). In our case, for a symmetric matrix, what we calculate is indeed equivalent to the pseudoinverse (or Moore-Penrose inverse), and so we have now adopted this more precise naming throughout.

Fig 7: Are the v4 values here the final ones, after bias correction? Can the values <0 be shown as well? Are there more, or less, in v4 than v3? The text says there are few <0 for v4 in a long term average, but as a user I'm more interested in the statistics of the daily data (otherwise I need to consider time-averaging the data prior to assimilation, and it's not clear how to time-average the AVKs). I see later in Fig 9 that v4 values < 0 are appreciable, so I think they should be shown here as well.

Yes, these are the final values. The values smaller than zero are in fact already shown. To make this clearer, both axes have been extended further. Note that it is the post filter that removes most of the large negative values, while it is by design that the small negatives are kept (by letting instrumental noise affect the column in both positive and negative directions, unbiased averages can be obtained, as explained in Whitburn et al., 2016 and Clarisse et al., 2019). As can be seen from the plot the number of negative values has not changed significantly between the two versions.

Fig 7: Showing a slope fit to only a subset of values is a bit odd. Perhaps show a reduced major axis fit, or just report the correlation?

The relation is not exactly linear, and a single slope from all data doesn't match the data visually. But we agree that it is a bit strange, and therefore removed the slope information from the graph. Instead,

we updated the range 15-20 % in the text to 10-20 % to encompass all HRI values (10% for the lowest HRI, 20% for the highest).

Fig 8: I suggest adding labels to the lat/lon lines, or removing them.  Otherwise I'm not sure what purpose they serve.

 We tried adding labels but found these too distracting. We clarified them instead in the caption: "Parallels are drawn every 15° and meridians every 30°."

464: Remove the comma after separately

This has been corrected.

467: errors —> error

This has been corrected.

481: without an

This has been corrected.

485: profiles,

This has been corrected.

510: Having participated in several such studies, I would say these uncertainties strike me as considerably underestimated.

Estimating uncertainties on the vertical profile is very difficult, as this really depends on local conditions. We will keep this comment in mind in future revisions of the product.

**Other changes**

In addition to the changes related to reviewer comments, we also fixed an error in equation (65) and made corrections in the acknowledgments.

---

## Author Response (AR2)

**RC1: 'Comment on amt-2023-48', Anonymous Referee #1, 26 Jun 2023**

The authors have done a good job of answering my questions and providing updates to the text and figures.

My only remaining significant comment is with regards to their response concerning an evaluation of NH3 trends in their updated retrieval. I agree it is complicated by trends in $NO_2$ and $SO_2$, though we also do have some knowledge of what the latter were. Further, there are trends in bottom up emissions inventories that, while uncertain, are credible on appropriate scales. It would certainly strengthen the paper to demonstrate that their improved trends are in alignment with these factors. But if not, I would urge them to at least explain (as they have done in the response text, partially), why such an analysis is not provided at this time.

Lastly, one small detail regarding the updated (thanks) Figure 3: The text mentions non-uniform values of the normalization factor off the east coast of North Africa — to be honest I'm not sure where that would be, do they mean the Red Sea? I don't see any issues there... Rather, it seems the factor is most often greater than one off the west coast of South Africa, South America, and perhaps the Himalayas as well.

I don't foresee the need for this article to undergo further review after the authors address these two issues.

We would like to thank the reviewer for checking the previous revision.

To properly assess trends would require relying either on the intervention of a model (to link columns to emissions and to take account of the trend changes of other inorganic species) or evaluation with FTIR total column measurements (which is already foreseen as a large follow-up study). In view of this, we prefer to go for the second option that the reviewer suggests "to explain why such an analysis is not provided at this time". We have expanded the paragraph on trends now as follows (at the very end of section 6):

*".... Apart from validation of the columns in an absolute or relative sense, comparison with FTIR columns will also allow evaluating regional $NH_3$ trends derived from IASI data. Such an evaluation could also be made with bottom-up inventories or with data derived from in-situ measured concentrations. However, in that case, there is the additional difficulty that long term trends of other inorganic pollutants ($NO_2$, $SO_2$) affect $NH_3$ columns differently than emissions or local concentrations (e.g. Lachatre et al., 2019), necessitating the intervention of a (chemistry transport) model."*

As for the second comment: thank you for this remark, we indeed meant to write "West coast of South Africa". This has now been corrected.